# SIRT1 Activator E1231 Alleviates Nonalcoholic Fatty Liver Disease by Regulating Lipid Metabolism

**Jiangxue Han [†], Shunwang Li [†], Weizhi Wang, Xinhai Jiang, Chao Liu, Lijuan Lei, Yining Li, Ren Sheng, Yuyan Zhang, Yexiang Wu, Jing Zhang, Yuhao Zhang, Yanni Xu * and Shuyi Si ***

NHC Key Laboratory of Biotechnology of Antibiotics, National Center for Screening Novel Microbial Drugs, Institute of Medicinal Biotechnology, Chinese Academy of Medical Sciences & Peking Union Medical College (CAMS & PUMC), Tiantan Xili 1#, Beijing 100050, China
* Correspondence: xuyanni2010@imb.pumc.edu.cn (Y.X.); sisy@imb.pumc.edu.cn (S.S.)
† These authors contributed equally to this work.

**Abstract:** Nonalcoholic fatty liver disease (NAFLD) is one of the most common liver diseases. Silencing information regulator 1 (SIRT1) was demonstrated to modulate cholesterol and lipid metabolism in NAFLD. Here, a novel SIRT1 activator, E1231, was studied for its potential improvement effects on NAFLD. C57BL/6J mice were fed a high-fat and high-cholesterol diet (HFHC) for 40 weeks to create a NAFLD mouse model, and E1231 was administered by oral gavage (50 mg/kg body weight, once/day) for 4 weeks. Liver-related plasma biochemistry parameter tests, Oil Red O staining, and hematoxylin-eosin staining results showed that E1231 treatment ameliorated plasma dyslipidemia, plasma marker levels of liver damage (alanine aminotransferase (ALT) and aspartate aminotransferase (AST)), liver total cholesterol (TC) and triglycerides (TG) contents, and obviously decreased hepatic steatosis score and NAFLD Activity Score (NAS) in the NAFLD mouse model. Western blot results showed that E1231 treatment significantly regulated lipid-metabolism-related protein expression. In particular, E1231 treatment increased SIRT1, PGC-1$\alpha$, and p-AMPK$\alpha$ protein expression but decreased ACC and SCD-1 protein expression. Additionally, in vitro studies demonstrated that E1231 inhibited lipid accumulation and improved mitochondrial function in free-fatty-acid-challenged hepatocytes, and required SIRT1 activation. In conclusion, this study illustrated that the SIRT1 activator E1231 alleviated HFHC-induced NAFLD development and improved liver injury by regulating the SIRT1-AMPK$\alpha$ pathway, and might be a promising candidate compound for NAFLD treatment.

**Keywords:** E1231; SIRT1; AMPK$\alpha$; FFA; NAFLD





## 1. Introduction

Nonalcoholic fatty liver disease (NAFLD), recently proposed to be replaced with the term metabolic-associated fatty liver disease, has become the most common chronic liver disease [1]. The prevalence of NAFLD is nearly 25% worldwide, and it is emerging as the most important and primary cause of liver-related mortality and end-stage liver disease [2,3]. Nonalcoholic fatty liver (NAFL) (characterized by 5% or greater hepatic steatosis with or without mild inflammation) and nonalcoholic steatohepatitis (NASH) (characterized by the presence of hepatocellular injury, hepatocyte ballooning, and inflammation) are two forms of NAFLD [1,3]. NAFL can develop into NASH, which has the potential to progress and result in liver fibrosis, cirrhosis, and hepatocellular carcinoma [4]. However, there are few therapeutic strategies to treat NAFLD other than changing lifestyle and managing associated complications [5]. Therefore, there is an urgent need to develop drugs for NAFLD treatment.

The pathogenesis of NAFLD is a complex dysmetabolic process, and much of its pathogenesis remains to be discovered [6,7]. The "two-hit" and "multiple-hit" hypotheses, which have been steadily complemented, propose that high concentrations of free fatty

acids (FFAs) induce hepatic steatosis as the first hit, which is a necessary factor for the occurrence of NAFLD [8]. Aberrant lipid accumulation within hepatocytes is a key factor for the progression of NAFLD. The imbalance between lipid intake and disposal in the liver leads to hepatocyte lipid accumulation, which creates a vicious cycle with hepatic lipotoxicity, insulin resistance, inflammation, oxidative stress, and mitochondrial dysfunction [9–11]. Regulating liver lipid metabolism is one of the most important strategies for NAFLD treatment.

Silencing information regulator 1 (SIRT1) is a member of the sirtuin protein family with nicotinamide adenine dinucleotide (NAD)$^+$-deacetylase activity [12], and it modulates hepatic energy metabolism through deacetylation of metabolic regulators [13]. Previous studies have demonstrated that SIRT1 may modulate cholesterol and lipid metabolism in NAFLD [14]. There is a decrease in liver SIRT1 expression in liver biopsies and in plasma from patients with NAFLD [15,16]. Liver-specific *Sirt1* knockout mice are prone to hepatic steatosis, endoplasmic reticulum stress, and liver inflammation; by contrast, overexpression of SIRT1 alleviates hepatic steatosis in high-fat-diet (HFD)-induced mice [17]. In addition, some activators of SIRT1 were shown to affect NAFLD by regulating lipid metabolism enzymes [18,19]. Therefore, SIRT1 is a potential therapeutic target for NAFLD. The development of SIRT1 activators would be a promising therapeutic strategy for NAFLD treatment.

In a previous study, we identified a compound E1231 as a novel SIRT1 activator and demonstrated that it improved lipid and cholesterol metabolism in hyperlipidemic golden hamsters and an Apolipoprotein E knockout (*ApoE*$^{-/-}$) mouse model of atherosclerosis [20]. However, the effect of E1231 on NAFLD is unclear. To identify whether it has an improvement effect on NAFLD, we examined its therapeutic potential and explored the mechanism of E1231 in high-fat and high-cholesterol diet (HFHC) induced NAFLD mice and free-fatty-acid (FFA)-challenged hepatocytes. Our study indicates that E1231 might be a promising candidate compound for NAFLD treatment.

## 2. Results

### 2.1. E1231 Treatment Ameliorated Plasma Biochemistry Parameter Abnormalities and Hepatic Steatosis in HFHC-Induced NAFLD Mice

Using the established homogeneous time-resolved fluorescence (HTRF) technology-based SIRT1 activator screening model [20], we first confirmed that E1231, a piperazine 1,4-diamide compound, could significantly activate SIRT1 in a dose-dependent manner with an EC$_{50}$ value of 0.31 μmol/L (Figure 1A). To evaluate the therapeutic effect of E1231 on NAFLD, we established a HFHC-induced mouse model of NAFLD. E1231 was administered by oral gavage at 50 mg/kg body weight once a day for 4 weeks to examine whether it had an improvement effect on NAFLD (Figure 1B).

Serum alanine aminotransferase (ALT) and aspartate aminotransferase (AST) are markers of levels of liver damage in NAFLD. Liver-related serum tests typically reflect a hepatocellular pattern of enzyme elevations for ALT and AST [4]. The plasma biochemistry analyses showed that the plasma ALT and AST in the HFHC group markedly increased when compared with the chow diet (CD) group (Figure 1C). However, E1231 treatment decreased the plasma levels of ALT ($p = 0.27$) and AST ($p = 0.17$) compared with the untreated HFHC group (Figure 1C). In addition, 40 weeks of HFHC feeding markedly increased the plasma levels of total cholesterol (TC), low-density lipoprotein cholesterol (LDL-C), and high-density lipoprotein cholesterol (HDL-C), and slightly increased the plasma level of triglycerides (TG) compared with the CD group (Figure 1D). Compared with the HFHC group, E1231 treatment significantly decreased the plasma levels of TC, TG, and LDL-C (Figure 1D), but obviously increased the plasma level of HDL-C (Figure 1D). These results indicated that E1231 treatment improved HFHC-induced dyslipidemia and liver injury.

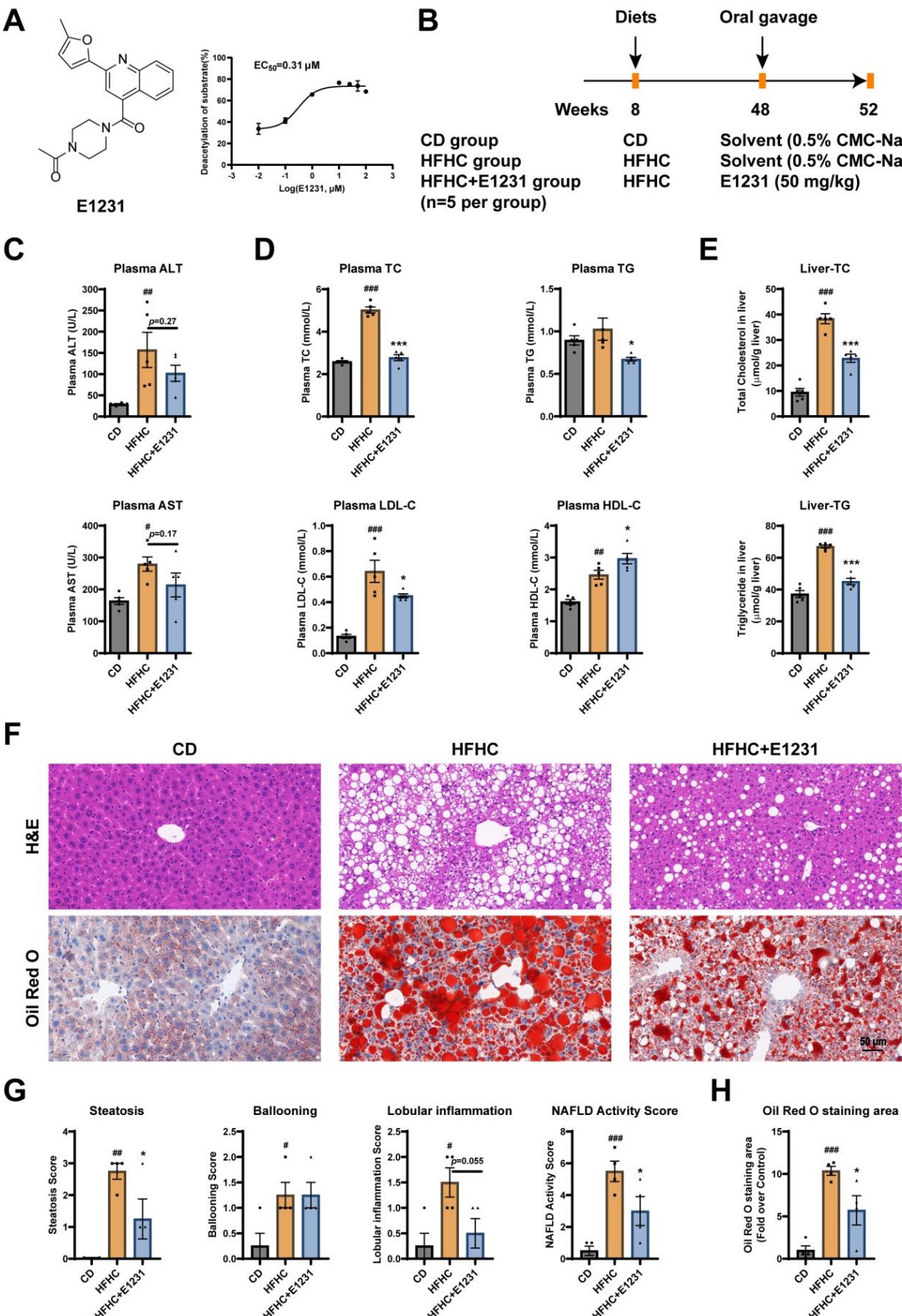

**Figure 1.** E1231 treatment alleviates NAFLD in a high-fat-and-high-cholesterol diet (HFHC) induced NAFLD mouse model. (**A**) The structure and activating SIRT1 activity of E1231. (**B**) Schedule of animal

experiments. Male C57BL/6J mice were randomly divided into three groups (Chow diet (CD) group, HFHC group, and HFHC + E1231 group), fed a normal chow diet (CD group) or a HFHC diet (HFHC group and HFHC + E1231 group) for 40 weeks. Then, E1231 (50 mg/kg body weight) was administered by oral gavage for 4 weeks in the HFHC + E1231 group; the same volume of solvent (0.5% CMC-Na) was given by oral gavage for 4 weeks in CD and HFHC groups. $n = 5$ per group. (**C**) Plasma alanine aminotransferase (ALT) and plasma aspartate aminotransferase (AST) levels. $n = 5$ per group. (**D**) Plasma total cholesterol (TC), triglycerides (TG), low-density lipoprotein cholesterol (LDL-C), and high-density lipoprotein cholesterol (HDL-C) levels. $n = 5$ per group. (**E**) Liver TC and TG contents (μmol/g liver) were determined. $n = 5$ per group. (**F**) Representative hematoxylin and eosin (H&E) and Oil Red O (ORO) stained images. (**G**) The steatosis score, lobular inflammation score, ballooning score, and NAFLD Activity Score (NAS) were calculated. $n = 4$ per group. (**H**) The quantification of ORO-staining images. $n = 4$ per group. The data are presented as the mean ± SEM. One-way ANOVA was used for analysis. HFHC group vs. CD group: # $p < 0.05$, ## $p < 0.01$, ### $p < 0.001$. HFHC + E1231 group vs. HFHC group: * $p < 0.05$, *** $p < 0.001$.

Next, the levels of TC and TG in livers were measured. As shown in Figure 1E, both liver TC and TG content in the HFHC group markedly increased compared with the CD group, while E1231 treatment significantly decreased TC and TG content in livers compared with the HFHC group. Hematoxylin and eosin (H&E) staining results showed that the histological structure of livers from the CD group was normal, while the livers from the HFHC group had severe steatosis, significant lipid-enriched vacuole formation, and inflammation, compared with the CD group (Figure 1F). Oil red O (ORO) staining results revealed that the livers from the HFHC group had much more lipid accumulation than the CD group, whereas histologic sections from the E1231 treatment group showed an obvious decrease in lipid accumulation compared with the HFHC group (Figure 1H). The NAFLD Activity Score (NAS) is a validated score used to grade disease activity in NAFLD patients [5,21]. The NAS is calculated by summing several component scores for steatosis (0–3), lobular inflammation (0–3), and ballooning (0–2) [22]. The NAS score of the CD group, HFHC group, and HFHC + E1231 group was about 0.5, 5.5, and 3.0, respectively (Figure 1G). Compared with the HFHC group, E1231 treatment decreased the steatosis ($p < 0.05$) and lobular inflammation ($p = 0.055$) scores but not the ballooning score (Figure 1G), and so significantly decreased the NAS score (Figure 1G). These data suggest that E1231 has an ameliorative effect on HFHC-induced NAFLD.

In addition, the effects of E1231 on liver inflammation were examined. Real-time quantitative polymerase chain reaction (RT-qPCR) results showed that the mRNA levels of inflammatory cytokines, including tumor necrosis factor α (*Tnfα*) ($p < 0.05$), interleukin-1β (*Il1β*) ($p = 0.12$), and interleukin-6 (*Il6*) ($p = 0.059$) in the livers of the HFHC group were higher than those in the CD group (Figure S1). E1231 treatment decreased the mRNA levels of *Tnfα* ($p = 0.065$), *Il1β* ($p = 0.14$), and *Il6* ($p = 0.068$) compared with the HFHC group (Figure S1).

In conclusion, our results indicated that E1231 treatment significantly improved HFHC-induced dyslipidemia and hepatic steatosis and inhibited HFHC-induced NAFLD development and liver injury.

### 2.2. E1231 Regulates SIRT1 and Other Lipid Metabolism Regulators in the Liver

Our data demonstrated that E1231 treatment significantly improved HFHC-induced dyslipidemia and hepatic steatosis, and that E1231 had an improvement effect against NAFLD. To elucidate how E1231 exerted an improvement effect against NAFLD, the mechanism was explored. The transcription factor sterol regulatory element-binding protein-1c (SREBP-1c) plays an important role in de novo lipogenesis (DNL). Additionally, the genes of fatty acid synthesis enzymes such as acetyl-coenzyme A carboxylase (ACC), fatty acid synthase (FAS), and stearoyl-CoA desaturase 1 (SCD-1), are the target genes of SREBP-1c [18,23]. SIRT1 regulates lipid metabolism by increasing levels of phosphorylated AMP-activated protein kinase alpha (p-AMPKα) [24], and decreasing the expression level

of downstream transcription factor SREBP-1c [25,26]. The SIRT1/PGC-1α (proliferator-activated receptor gamma coactivator 1α) pathway was reported to regulate mitochondrial physiology and lipid autophagy in NAFLD [27–30]. In addition, peroxisome proliferator-activated receptor gamma 2 (PPARγ2) activation plays a major role in high-fat-diet (HFD)-induced fatty liver development [31]. Therefore, to illustrate the effects of E1231 treatment on HFHC-induced NAFLD, we performed Western blots or RT-qPCR to examine the expression levels of these key regulatory proteins related to lipid metabolism.

As shown in Figure 2, Western blot analysis revealed that HFHC feeding significantly decreased the protein expression levels of SIRT1, p-AMPKα/AMPKα ratio, and PCG-1α (Figure 2A,B) but dramatically increased the protein expression levels of lipogenic proteins, including SREBP-1c, ACC, and SCD-1, when compared with the control group fed on a chow diet (Figure 2C,D). E1231 treatment significantly increased the protein expression levels of SIRT1 and PCG-1α, and the p-AMPKα/AMPKα ratio (Figure 2A,B) but obviously decreased the expression levels of SREBP-1c, ACC, and SCD-1 when compared with the HFHC group (Figure 2C,D). RT-qPCR results showed that the mRNA expression levels of *Srebp1c*, *Scd1*, and *Fas* were significantly increased in HFHC-fed mice compared with CD-fed mice. The mRNA expression levels of *Acc* increased ($p = 0.074$) in HFHC-fed mice compared with CD-fed mice, indicating that there might be an increasing trend that did not reach statistical significance. E1231 treatment significantly decreased the mRNA expression levels of *Scd1* and *Fas* when compared with those in HFHC-fed mice (Figure 2E). Additionally, E1231 treatment also decreased the mRNA expression levels of *Srebp1c* ($p = 0.11$) and *Acc* ($p = 0.33$) (Figure 2E) but with no statistical significance. In addition, the mRNA expression levels of *Pparγ2* ($p = 0.11$) and its target gene scavenger receptor *Cd36* ($p = 0.053$) had an increasing trend that did not reach statistical significance in HFHC-fed mice compared with CD-fed mice. Additionally, another *Pparγ2* target gene adipose differentiation-related protein (*Adrp*) was significantly increased in HFHC-fed mice compared with CD-fed mice, while *Pparγ1* mRNA levels were not altered (Figure 2F). E1231 treatment significantly decreased the mRNA expression of *Pparγ2* and *Adrp* (Figure 2F), and it also decreased the mRNA expression of *Pparγ1* and *Cd36* which did not reach statistical significance (Figure 2F). It was concluded that E1231 ameliorated HFHC-induced NAFLD development by increasing the activities of SIRT1 and p-AMPKα and decreasing the key lipogenesis-related proteins, including SREBP-1c, ACC, SCD-1, and PPARγ2.

*2.3. E1231 Suppresses FFA-Induced Lipid Accumulation in the Human Hepatoma Cell Line HepG2*

An overwhelming influx of FFAs causes hepatotoxicity. Because hepatocytes are the major cells responsible for FFA-challenged lipotoxicity [32,33], we investigated the direct effects of E1231 on lipid accumulation in FFA-challenged hepatocytes. FFA (oleic acid (OA): palmitic acid (PA) = 2:1, *V/V*) was used to induce a well-established steatosis model [34]. ORO staining and cellular lipid content analysis showed that FFAs significantly increased lipid accumulation in HepG2 cells in a dose-dependent manner (Figure 3A), with more than 2-fold at 500 and 1000 μM compared with control cells (Figure 3A). We then chose to examine the effects of E1231 on lipid accumulation in 500 μM FFA-challenged hepatocytes. As shown in Figure 3B,C, ORO and Nile red fluorescent staining results showed that E1231 treatment greatly decreased the number of lipid droplets at concentrations of 0.1 and 1 μM compared with the FFA-only group. In addition, cellular lipid content analysis results showed that E1231 treatment significantly decreased the TG content in FFA-challenged cells compared with the FFA-only group (Figure 3D).

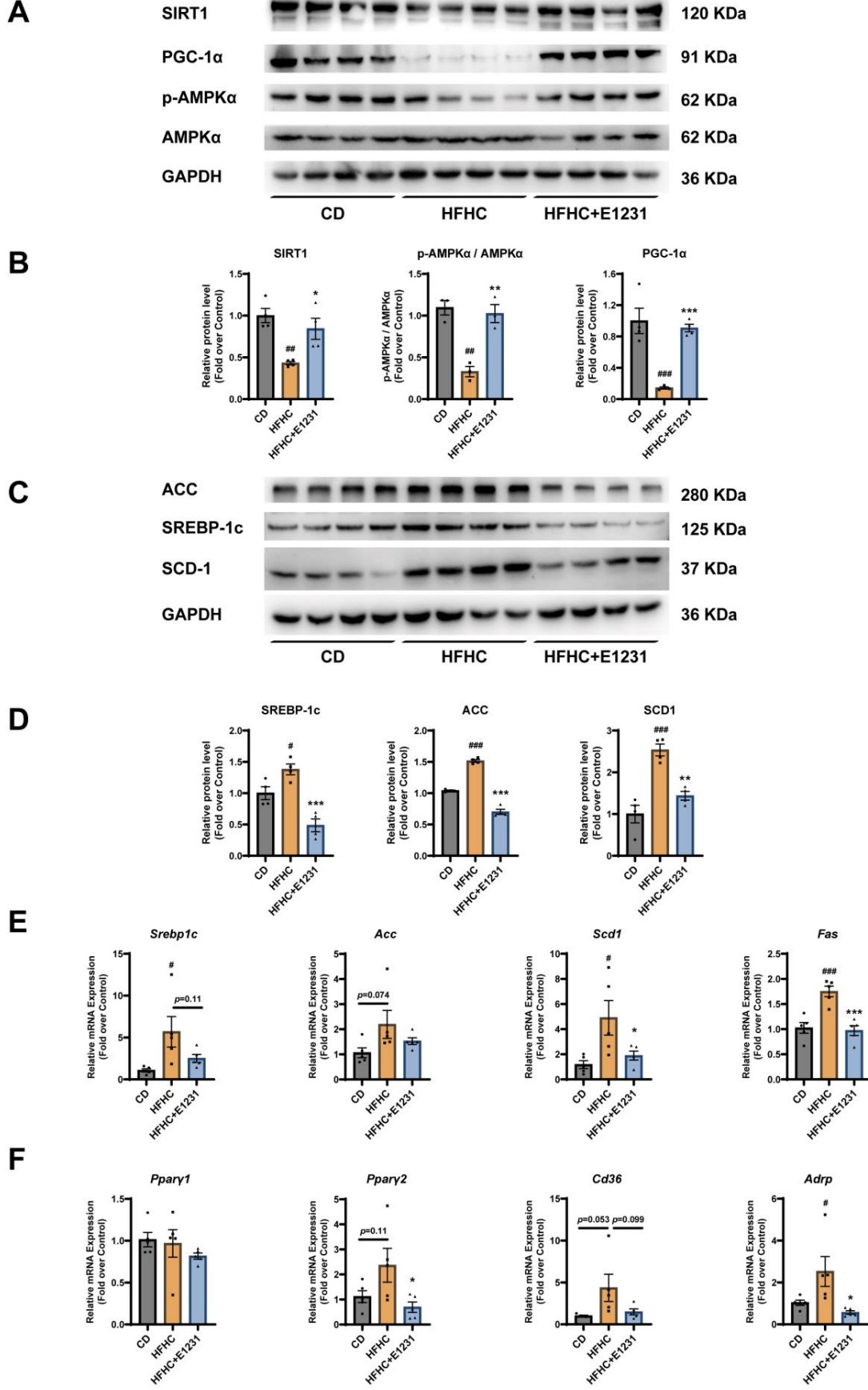

**Figure 2.** E1231 regulates silencing information regulator 1 (SIRT1) and other lipid metabolism regulators in the liver of NAFLD mice. (**A**,**C**) Western blot analysis of lipid metabolism regulators (including

SIRT1, phosphorylated AMP-activated protein kinase alpha (p-AMPKα), AMP-activated protein kinase alpha (AMPKα), acetyl-coenzyme A carboxylase (ACC), sterol regulatory element-binding protein-1c (SREBP-1c), stearoyl-CoA desaturase 1 (SCD-1), and peroxisome proliferator-activated receptor gamma coactivator 1α (PGC-1α)) in the livers of the indicated mice. Glyceraldehyde 3-phosphate dehydrogenase (GAPDH) was used as a control. (**B,D**) Densitometry of the band intensity (ratio to GAPDH) was performed. Data are presented as the mean ± SEM. $n$ = 4 per group. (**E**) Real-time–quantitative polymerase chain reaction (RT-qPCR) analysis of the liver mRNA expression levels of sterol regulatory element-binding protein-1c (*Srebp1c*), acetyl-coenzyme A carboxylase (*Acc*), fatty acid synthase (*Fas*), and stearoyl-CoA desaturase 1 (*Scd1*). $n$ = 5 per group. (**F**) RT-qPCR analysis of the liver mRNA expression levels of peroxisome proliferator-activated receptor gamma 2 (*Pparγ2*), *Pparγ1*, scavenger receptor *Cd36*, and adipose differentiation-related protein (*Adrp*). $n$ = 5 per group. The data are presented as the mean ± SEM. One-way ANOVA was used for analysis. HFHC group vs. CD group: # $p < 0.05$, ## $p < 0.01$, ### $p < 0.001$. HFHC + E1231 group vs. HFHC group: * $p < 0.05$, ** $p < 0.01$, *** $p < 0.001$.

To investigate the molecular regulation mechanism by E1231, we performed Western blot analysis on SIRT1 and its downstream regulators in lipid metabolism (p-AMPKα, AMPKα, ACC) in HepG2 cells. As shown in Figure 3E, E1231 at concentrations of 1 μM significantly increased the protein expression of SIRT1 and the p-AMPKα/AMPKα ratio compared with the FFA-only group, which was in accordance with the in vivo results. Furthermore, E1231 treatment at 1 μM clearly decreased the protein level of ACC caused by FFAs (Figure 3E). Taken together, our data indicated that E1231 inhibited FFA-induced lipid accumulation in HepG2 cells by regulating the SIRT1-AMPKα pathway.

## 2.4. E1231 Suppresses FFA-Challenged Lipid Accumulation in Alpha Mouse Liver 12 (AML12) Cells

We then evaluated the lipid metabolism-regulating effect of E1231 in FFA-challenged AML12 cells. Consistent with the HepG2 cells, FFAs significantly increased the lipid accumulation compared with DMSO-treated control cells in a dose-dependent manner according to the ORO staining results (Figure 4A). ORO staining and Nile red fluorescent staining results showed that E1231 treatment clearly decreased lipid accumulation in AML12 cells challenged with 500 μM FFA in a dose-dependent manner (Figure 4B,C). Analysis of the cellular lipid content revealed that exposure to FFAs increased the amount of cellular TG, whereas treatment with E1231 clearly reduced the amount of cellular TG relative to the FFA group (Figure 4D). Collectively, these data indicate that E1231 suppressed lipid accumulation in FFA-challenged hepatocytes.

## 2.5. E1231 Represses Lipid Accumulation in FFA-Challenged Hepatocytes by Activating SIRT1

We demonstrated that E1231 was a SIRT1 activator in a previous study [20] and in this work (Figure 1A). SIRT1 has been reported to protect against NAFLD through the regulation of lipid homeostasis [14,35]. EX527 is a potent and selective SIRT1 inhibitor [36,37]. Man Li et al. showed that the inhibition of SIRT1 by EX527 could markedly enhance FFA-induced lipid accumulation [38]. Therefore, to investigate whether the regulation of lipid metabolism and thus the anti-NAFLD effect of E1231 was mediated by SIRT1, we added EX527 to the FFA-challenged HepG2 and AML12 cell lipid accumulation model.

As shown in Figure 5, EX527 clearly reversed the inhibitory effect of E1231 on lipid deposition in both HepG2 and AML12 cells as shown by ORO staining (Figure 5A,B) and Nile red fluorescent staining results (Figure 5C,D). In addition, EX527 reversed the inhibitory effect of E1231 on TG accumulation in both HepG2 and AML12 cells, as demonstrated by a cellular lipid content analysis (Figure 5E). Western blot experiments were then performed to investigate the potential mechanism in both HepG2 and AML12 cells when treated with or without E1231 and EX527, respectively. The results showed that the increased SIRT1 protein expression effect induced by E1231 was obviously blunted when it co-existed with EX527 in both HepG2 and AML12 cells (Figure 5F,G). Collectively, these data indicate that E1231 inhibited lipid accumulation in FFA-challenged hepatocytes by activating SIRT1.

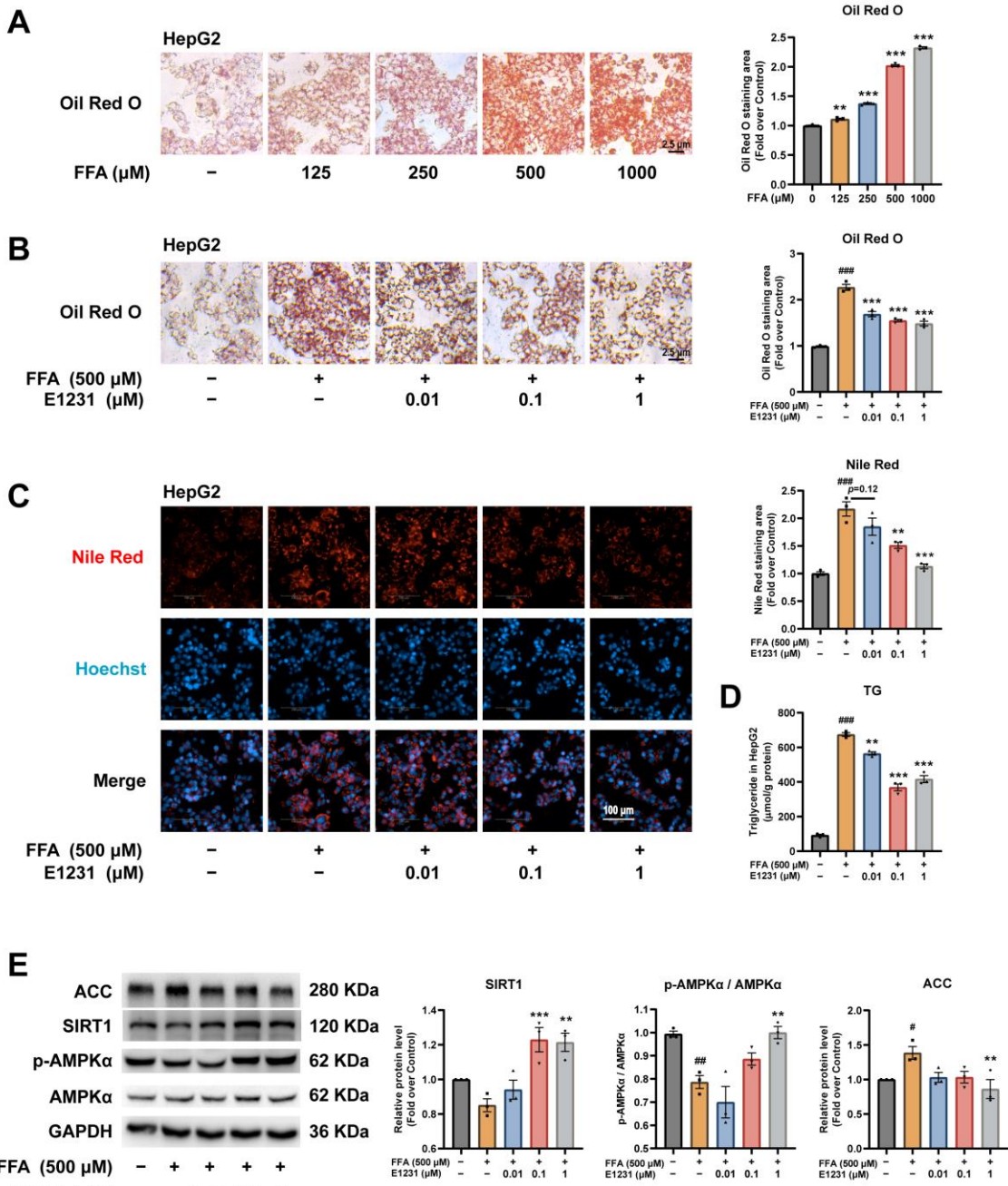

**Figure 3.** E1231 reduces lipid accumulation in FFA-challenged human hepatoma cell line HepG2 cells. (**A**) ORO staining and quantification of lipid accumulation in FFA-challenged HepG2 cells. HepG2 cells were treated with free fatty acid (FFA, oleic acid (OA): palmitic acid (PA) = 2:1, *V*/*V*) at 125, 250, 500, and 1000 μM for 24 h to generate the HepG2 in vitro steatosis cell model. Scale bar, 2.5 μm. (**B**–**D**) ORO staining and quantification (**B**), Nile red staining and quantification (**C**), and TG contents (**D**) in FFA (500 μM)-challenged HepG2 cells treated with dimethylsulfoxide (DMSO) or E1231 (0.01, 0.1, and 1 μM) for 24 h. Scale bar for (**B**), 2.5 μm. Scale bar for (**C**), 100 μm. (**E**) Western blot analysis of lipid metabolism regulator proteins (SIRT1, p-AMPKα, AMPKα, ACC) in FFA-challenged HepG2 cells treated with E1231 or DMSO. GAPDH was used as a control. Densitometry of the band intensity (ratio to GAPDH) was performed. Data from 3 independent experiments are presented as the mean ± SEM. One-way ANOVA was used for analysis. (**A**) FFA-only group vs. control (DMSO) group: ** $p < 0.01$, *** $p < 0.001$. (**B**–**E**) FFA-only group vs. control (DMSO) group: # $p < 0.05$, ## $p < 0.01$, ### $p < 0.001$. E1231 group vs. FFA-only group: ** $p < 0.01$, *** $p < 0.001$.

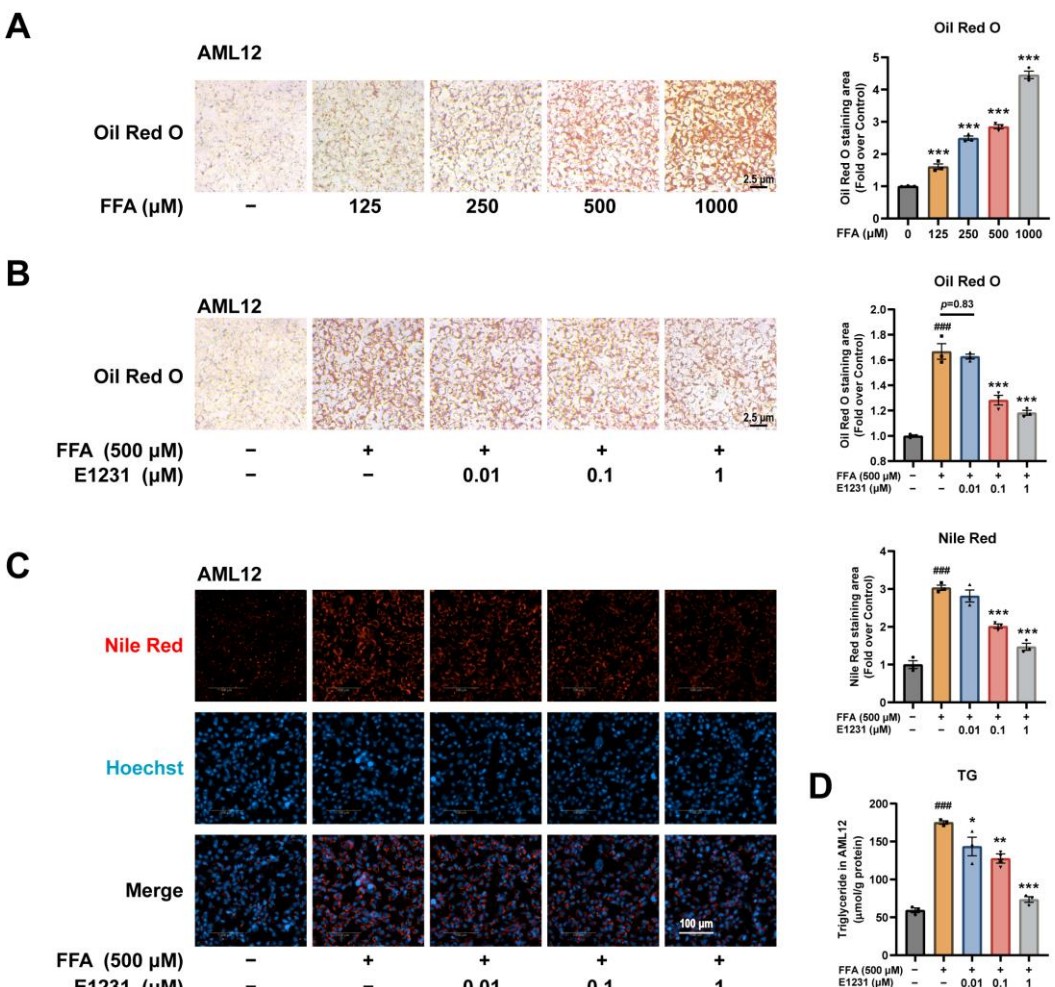

**Figure 4.** E1231 reduces lipid accumulation in FFA-challenged alpha mouse liver 12 (AML12) cells. (**A**) ORO staining and quantification of lipid accumulation in FFA-challenged AML12 cells. AML12 cells were treated with FFA (OA: PA = 2:1, *V/V*) at 125, 250, 500, and 1000 μM for 24 h to generate the in vitro lipid accumulation cell model. Scale bar, 2.5 μm. (**B–D**) ORO staining and quantification (**B**), Nile red staining and quantification (**C**), and TG contents (**D**) in FFA (500 μM)-challenged HepG2 cells treated with DMSO or E1231 (0.01, 0.1, and 1 μM) for 24 h. Scale bar for (**B**), 2.5 μm. Scale bar for (**C**), 100 μm. Data from 3 independent experiments are presented as the mean ± SEM. One-way ANOVA was used for analysis. (**A**) FFA-only group vs. control group: *** $p < 0.001$. (**B–D**) FFA-only group vs. control (DMSO) group: ### $p < 0.001$. E1231 group vs. FFA-only group: * $p < 0.05$, ** $p < 0.01$, *** $p < 0.001$.

### 2.6. E1231 Improves Mitochondrial Function in FFA-Challenged Hepatocytes by Activating SIRT1

Chronic impairment of lipid metabolism leads to alterations in the oxidant/antioxidant balance, cellular lipotoxicity, lipid peroxidation, chronic endoplasmic reticulum (ER) stress, and mitochondrial dysfunction [9,39]. Malondialdehyde (MDA) is a marker of oxidative-stress-induced lipid peroxidation products. SIRT1, PGC-1α, and AMPKα are the major regulators of mitochondrial biogenesis and function [40]. Our data indicated that E1231 treatment increased SIRT1, PGC-1α, and p-AMPKα expression. Therefore, we monitored the MDA contents in the livers and the mitochondrial function in FFA-challenged hepatocytes.

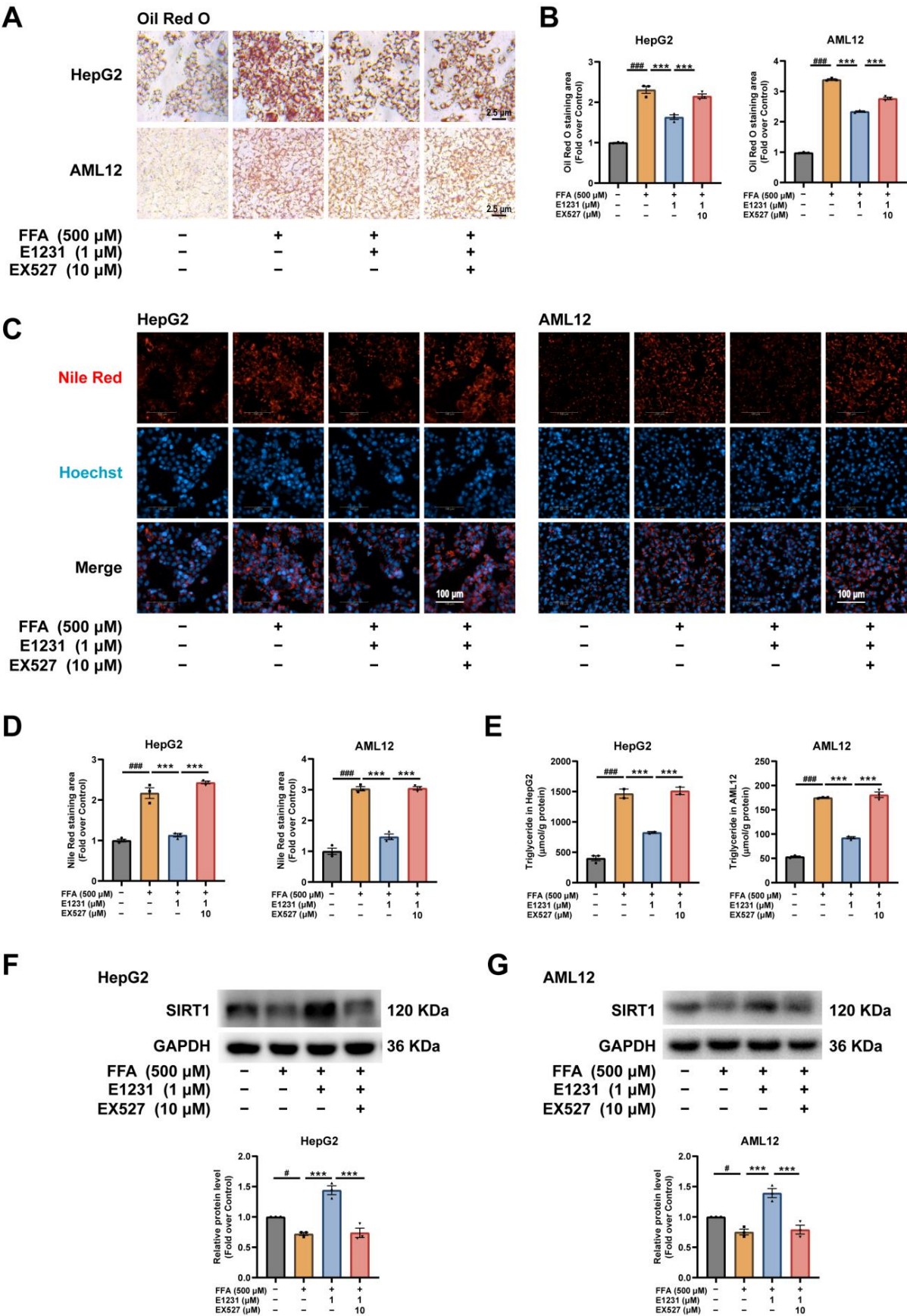

**Figure 5.** E1231 reduced lipid accumulation in FFA-challenged hepatocytes by activating SIRT1. HepG2 and AML12 cells were treated with FFA at 500 μM for 24 h, and then DMSO, E1231 (1 μM),

or EX527 was added for another 24 h. (**A**–**D**) ORO staining (**A**,**B**) and Nile red fluorescent staining (**C**,**D**) of lipid accumulation in FFA-challenged HepG2 and AML12 cells treated with DSMO, E1231, or EX527. Scale bar for (**A**), 2.5 μm. Scale bar for (**C**), 100 μm. (**E**) TG content of lipid accumulation in FFA-challenged HepG2 and AML12 cells treated with DSMO, E1231, or EX527. (**F**,**G**) Western blot analysis of SIRT1 in FFA-challenged HepG2 cells treated with DSMO, E1231, or EX527. Densitometry of the band intensity (ratio to GAPDH) was performed. The data from 3 independent experiments are presented as the mean ± SEM. One-way ANOVA was used for analysis. FFA-only group vs. control (DMSO) group: # $p < 0.05$, ### $p < 0.001$. E1231 group vs. FFA-only group or E1231 and EX527 group: *** $p < 0.001$.

As shown in Figure S2, HFHC feeding significantly increased the liver MDA contents compared with the CD feeding group, whereas E1231 treatment obviously decreased the liver MDA contents compared with the HFHC group (Figure S2). Tetramethylrhodamine ethyl ester (TMRE), a key indicator of cell health, was used to monitor the changes in the mitochondrial membrane potential (MMP). The results showed that FFA treatment resulted in mitochondrial depolarization compared with DMSO-treated control cells, while E1231 at 0.1 and 1 μM could obviously normalize the MMP depolarization induced by FFA in both HepG2 and AML12 cells (Figure 6A,B). The protective effect of E1231 on FFA-challenged MMP depolarization in both HepG2 and AML12 cells was markedly suppressed by EX527 (Figure 6C,D). These data indicated that E1231 could attenuate chronic lipid oversupply-induced oxidative stress and FFA-induced mitochondrial dysfunction in a SIRT1-dependent manner.

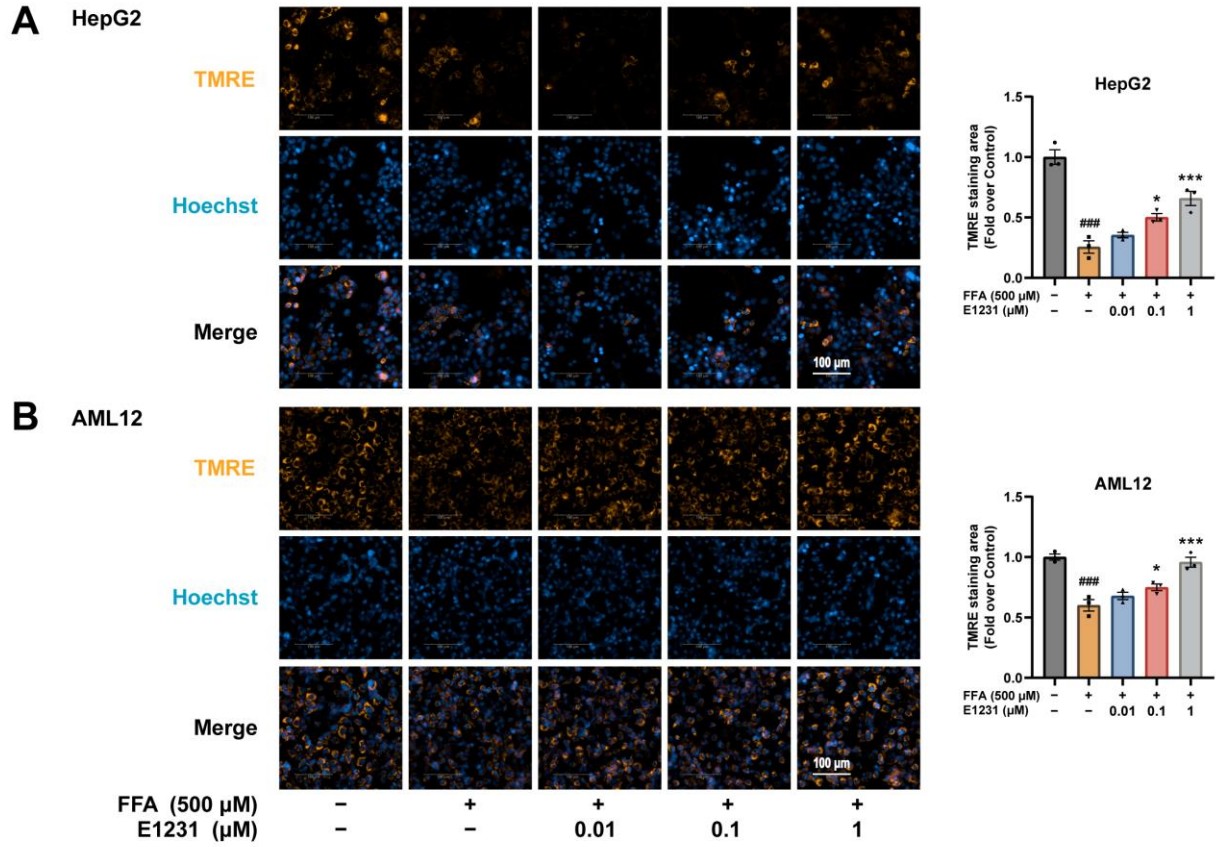

**Figure 6.** *Cont.*

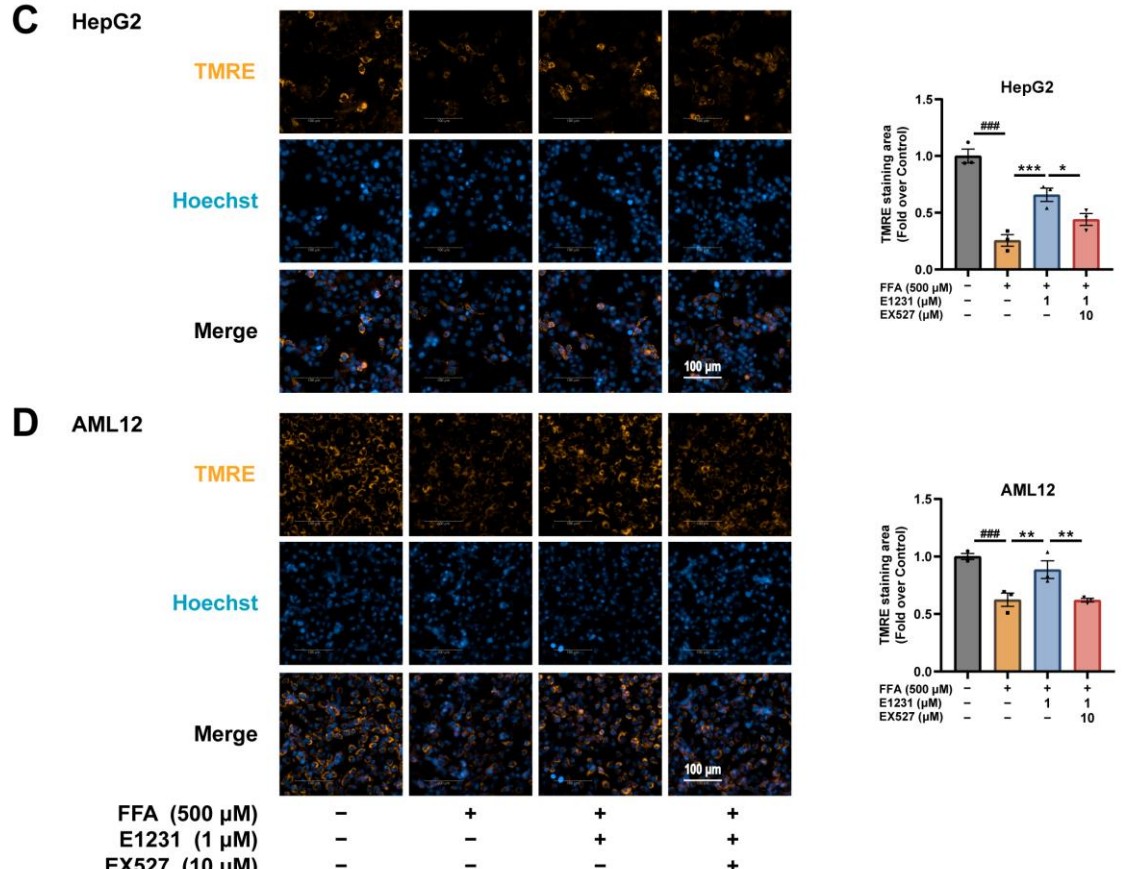

**Figure 6.** E1231 improves mitochondrial function in FFA-challenged HepG2 and AML12 cells. HepG2 and AML12 cells were treated with FFA at 500 μM for 24 h, and then DMSO, E1231, or EX527 at the indicated concentrations was added for another 24 h. (**A**,**B**) Tetramethylrhodamine ethyl ester (TMRE) staining of lipid accumulations in FFA-challenged HepG2 (**A**) and AML12 (**B**) cells treated with DMSO or E1231 (0.01, 0.1, and 1 μM). Scale bar, 100 μm. (**C**,**D**) TMRE-mediated MMP staining of lipid accumulation in FFA-challenged HepG2 (**C**) and AML12 (**D**) cells treated with DSMO, E1231, or EX527. Scale bar, 100 μm. The representative images are from three independent experiments. The data are presented as the mean ± SEM. One-way ANOVA was used for analysis. FFA-only group vs. control (DMSO) group: ### $p < 0.001$. E1231 group vs. FFA-only group or E1231 and EX527 group: * $p < 0.05$, ** $p < 0.01$, *** $p < 0.001$.

## 3. Discussion

NAFLD is a common chronic liver disease worldwide and is associated with metabolic syndrome. Hepatic steatosis is the hallmark of NAFLD, and it results from the imbalance between lipid synthesis and lipid utilization. The dysregulated lipid metabolism in hepatocytes creates a lipotoxic environment, promoting the development of NAFLD [41]. Enhanced SREBP1c-mediated DNL contributes significantly to the intrahepatic accumulation of lipids in NAFLD [42,43]. SIRT1 may ameliorate NAFLD and plays important roles in reducing hepatic steatosis and regulating mitochondrial function involved in the process of NAFLD [27,28,44,45]. SIRT1 could increase AMPKα activity and inhibit SREBP1c activity, and thus decrease lipogenesis in mouse livers [45,46]. E1231 is a novel SIRT1 activator previously identified by our laboratory [20]. In this study, we uncovered the first evidence that E1231 treatment significantly improved liver steatosis in HFHC-induced NAFLD mice and decreased lipid accumulation in FFA-challenged hepatocytes. The mechanistic studies showed that E1231-regulated lipid metabolism by regulating the SIRT1-AMPKα pathway and decreasing the downstream key molecules of lipogenesis, including SREBP-1c, ACC, SCD-1, and PPARγ2 (Figure 7). These data indicate that SIRT1 activator E1231 ameliorated

lipid accumulation and thus protected against NAFLD in both FFA-challenged hepatocytes and HFHC-induced NAFLD mice.

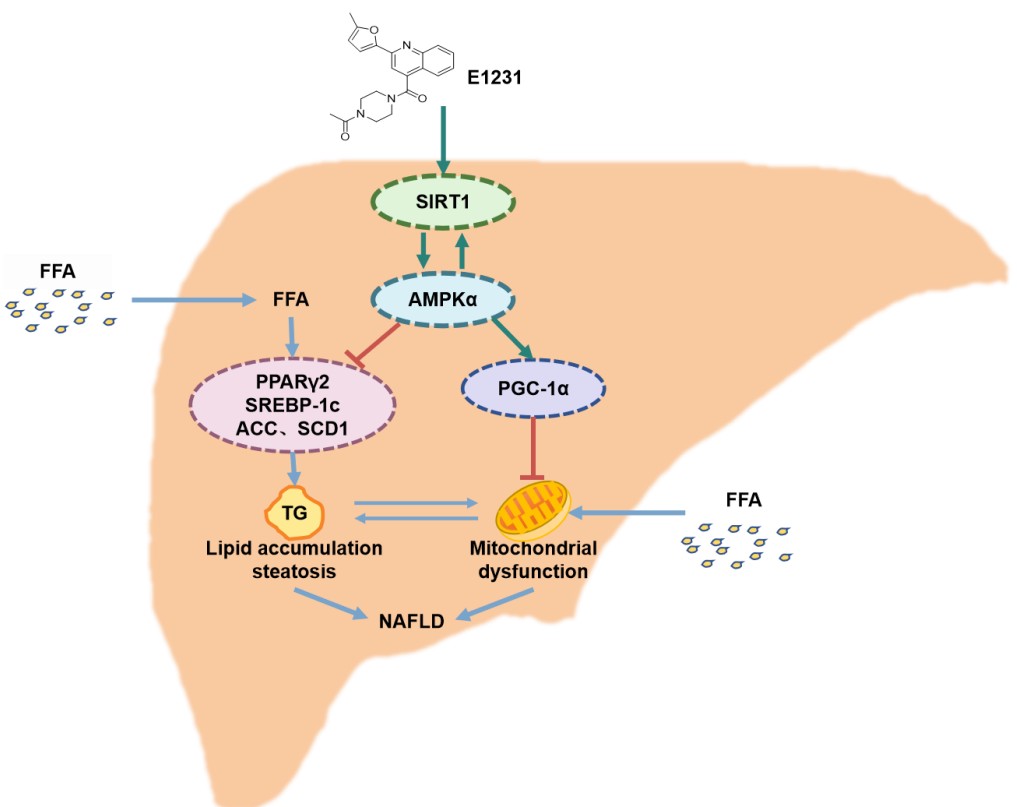

**Figure 7.** A schematic mechanism diagram of E1231's inhibitory effect on NAFLD. Specifically, E1231 increases the activity of SIRT1-AMPKα but decreases the expression of the key lipogenesis proteins SREBP-1c, ACC, SCD-1, and PPARγ2. As a result, E1231 decreases lipid accumulation and improves steatosis and FFA-induced mitochondrial dysfunction, and then exerts an improvement effect on NAFLD. SIRT1, silencing information regulator 1; AMPKα, AMP-activated protein kinase; FFA, free fatty acid; SREBP-1c, sterol regulatory element-binding protein-1c; ACC, acetyl-coenzyme A carboxylase; SCD-1, stearoyl-CoA desaturase 1; PPARγ2, peroxisome proliferator-activated receptor gamma 2; NAFLD, nonalcoholic fatty liver disease.

The dysregulation of lipid homeostasis in hepatocytes could generate toxic lipids that result in dysfunctional organelles promoting inflammation, hepatocellular damage, oxidative stress, mitochondrial dysfunction, and cell demise [10]. Mitochondria constitute the principal site for oxidative phosphorylation (OXPHOS) and fatty acid oxidation (FAO) (or β-oxidation), and mitochondrial dysfunction leads to the increased accumulation of FAO and TG in hepatocytes [47]. SIRT1 and AMPKα regulate mitochondrial function through the activation of PGC-1α, which is a master regulator of mitochondrial function. Our results showed that E1231 could inhibit liver inflammation, decrease lipid peroxidation product MDA content, and improve mitochondrial function. In addition, there are many signaling cross-talks between the liver and white adipose tissue (WAT) [48], and SIRT1 activity can affect WAT metabolism and then have an impact on liver metabolism through metabolic substrate delivery or altered inflammatory signaling [49]. Therefore, further studies need to be conducted to investigate the effects of E1231 on WAT SIRT1 activity and WAT metabolism. Finally, since there are many NAFLD models (such as animal models induced by a high-fat diet, high-fat high-fructose diet, and methionine-choline-deficient diet), further studies need to be conducted to investigate the comprehensive effects of E1231 on NASH and liver fibrosis.

SIRT1 is an NAD$^+$-dependent deacetylase [50], and activation of SIRT1 has multiple biological functions that are beneficial for life extension, including the improvement in metabolic diseases such as NAFLD, diabetes, and obesity [14,51]. A variety of SIRT1-activated compounds (STACs), including natural and synthetic STACs (e.g., SRT1720, SRT2104, and SRT3025), have been identified [52]. SRT1720 treatment directly reduced the expression of lipogenic genes, such as SREBP-1c, ACC, and FAS, reduced liver lipid accumulation, and reduced oxidative stress and inflammation, which improved the development of NAFLD in monosodium-glutamate (MSG)-induced diabetic mice [53]. Our study demonstrated that the SIRT1 activator E1231 could alleviate NAFLD development, which adds new evidence for SIRT1 activators being used for NAFLD treatment. Therefore, the development of SIRT1 activators may become a promising therapeutic strategy for NAFLD [13,35,54].

## 4. Materials and Methods

### 4.1. Establishment of an HFHC-Induced Mouse Model of NAFLD

Eight-week-old male C57BL/6J mice were purchased from Huafukang Bioscience (Beijing, China). All animals were treated in accordance with the Guide for the Care and Use of Laboratory Animals of the Institute of Medicinal Biotechnology, CAMS & PUMC. All animal experiments were approved by the Animal Care and Use Committee of the Institute of Medicinal Biotechnology, CAMS & PUMC.

The mice were divided into three groups: the CD group, HFHC group, and HFHC + E1231 group. Each group had five mice. The mice in the control group were fed a chow diet (CD, #Co60, SPF; Beijing Biotechnology Co., Ltd., Beijing, China). The mice in the model group and E1231 group were fed a HFHC diet (#M10640, 40 kcal% fat, 20 kcal% fructose, and 2% cholesterol, BIOPIKE, Beijing, China) for 40 weeks to establish a HFHC-induced NAFLD model. Then, the E1231-treated group was administered E1231 (J & K Scientific, Beijing, China), 50 mg/kg in 0.5% sodium carboxyl methyl cellulose (CMC-Na, #C9481, CAS:9004-32-4, Sigma-Aldrich, St. Louis, MO, USA) once daily by oral gavage for 4 weeks, and the CD group and HFHC group mice were administered 0.5% CMC-Na for 4 weeks.

### 4.2. Plasma Assay and Hepatic Lipid Assay

The plasma ALT, AST, TC, HDL-C, LDL-C, and TG levels were measured using a fully automated biochemical analyzer (Hitachi 71800, Chiyoda, Japan) and commercially available assay kits (BIOSINO Biotechnology, Beijing, China).

Liver TC and TG contents were briefly determined as follows [55]. Liver tissues were weighed and homogenized. Then, the hepatic lipid was isolated by an extract liquor (methanol:chloroform = 2:1, $V/V$). The solvent was evaporated using a vacuum centrifugal concentrator and the lipids were dissolved in 10% Triton X-100 solution. The TC and TG in hepatic lipids were then measured (#E1015 for TC, #E1013 for TG, APPLYGEN, Beijing, China) according to the manufacturer's protocols using a Multimode Plate Reader (EnVision 2105, PerkinElmer, Fremont, CA, USA). In addition, the cellular TG content was measured with a commercial kit (#E1013, APPLYGEN, Beijing, China). The TC and TG contents in the livers were calculated as μmol/g liver weight, respectively.

### 4.3. Histological Analysis

The liver tissues were fixed in a 4% paraformaldehyde solution (#BL539A, Biosharp, Hefei, China) and then dehydrated in 20% sucrose. Then, the tissues were embedded in paraffin or O.C.T. Compound (#4583, SAKURA Tissue-Tek, Torrance, CA, USA) and mounted on slides. The paraffin tissue slides were subjected to H&E staining to evaluate the histological structure of the livers. The frozen samples were sectioned and stained with ORO to visualize lipid droplets. Liver sections were stained with ORO solution (#H8070, Solarbio, Beijing, China) and Mayer's hematoxylin solution (#G1260, Solarbio, Beijing, China). H&E-stained images were scanned using 3DHISTECH's Slide Converter. The ORO-

stained images were taken under a light microscope (Leica DMIL, Leica Microsystems, Wetzlar, Germany).

### 4.4. Cell Culture

HepG2 cells and AML12 cells were obtained from ATCC (Rockville, MD). Cells were cultured in Dulbecco's modified Eagle medium (DMEM) (#C11995500BT, Gibco, Billings, NK, USA) supplemented with 10% fetal bovine serum (FBS) (#10270-106, Gibco, Billings, NK, USA) ($V/V$) under conditions of 37 °C and 5% $CO_2$ in an incubator.

### 4.5. FFA-Challenged Lipid Accumulation Model in Hepatocytes

PA (#10006627, Cayman Chemical, Ann Arbor, MI, USA) and OA (#373768, CAS: 143-19-1, J & K Scientific, Beijing, China) were purchased from Sigma-Aldrich (St. Louis, MO, USA). HepG2 and AML12 cells were treated with 125, 250, 500, and 1000 μM FFA (OA:PA = 2:1, $V/V$) and 0.5% fatty acid-free bovine serum albumin (#BAH68, Equitech Bio, Kerrville, TX, USA) for 24 h. Then, ORO (#G1262, Solarbio, Beijing, China) staining was performed to determine the level of lipid accumulation in the cells. Five ORO-stained images per sample were taken under a light microscope (Leica DMIL, Leica Microsystems, Wetzlar, German), and then the density of lipid droplets was analyzed by ImageJ software (National Institutes of Health, Bethesda, MD, USA). According to the results, FFA at 500 μM was used to perform the following experiments.

### 4.6. E1231 Treatment in FFA-Challenged Hepatocytes

To examine the effect of E1231 on lipid accumulation, HepG2 and AML12 cells were treated as follows. HepG2 and AML12 cells were first incubated with 500 μM FFA for 24 h, and then E1231 at the indicated concentration (0, 0.01, 0.1, and 1 μM) was added and cells were incubated for another 24 h. To determine whether the effect of E1231 on lipid accumulation was dependent on its activation of SIRT1, HepG2 and AML12 cells were treated with 500 μM FFA with or without E1231 (1 μM), and the SIRT1 inhibitor EX527 (#S1541, Selleck Chem, Houston, TX, USA) (10 μM), for 24 h. Then, the cells were stained with ORO, Nile red, or TMRE according to the instructions of commercial reagents or kits or analyzed by Western blot.

### 4.7. Nile Red Fluorescent Staining

Nile red fluorescent staining reagent (#C0009, Applygen, Beijing, China) was used as follows. HepG2 and AML12 cells were treated as above with FFAs and E1231 with or without EX527. The cells were then fixed with a 4% paraformaldehyde solution for 10 min and then washed with phosphate-buffered saline (PBS) three times. Cells were stained with Nile red staining solution (10 μM) for 10 min. Hoechst (#R37605, Hoechst 33342, Thermo, Waltham, MA, USA) was used to stain the cell nuclei.

### 4.8. Tetramethylrhodamine Ethyl Ester (TMRE) Staining

A TMRE-Mitochondrial Membrane Potential Assay kit (#T669, Invitrogen, Carlsbad, CA, USA) was used for quantifying changes in the mitochondrial membrane potential in live cells. HepG2 and AML12 cells were stained with TMRE solution (200 nM) for 20 min. Hoechst (#R37605, Hoechst 33342, Thermo, Waltham, MA, USA) was used to stain the cell nuclei.

The fluorescent stained images were captured and analyzed by a High Content Analysis System (Operetta CLS, PerkinElmer, Waltham, MA, USA) (for Nile red, Ex: 488 nm, Em: 528 nm; for TRME, Ex: 549 nm, Em: 574 nm; for Hoechst, Ex: 360 nm, Em: 460 nm).

### 4.9. Western Blot Analysis

Total liver tissue and hepatocyte proteins were extracted by RIPA lysis buffer (#C1053, APPLGEN, Beijing, China) containing protease inhibitor (#04693132001, Roche, Basel, Switzerland). The protein concentration was quantified by a BCA assay kit (#23225, Thermo,

Waltham, MA, USA). The protein samples were separated by 10% SDS-PAGE and then transferred to polyvinylidene fluoride (PVDF) membranes (IPVH00010; Millipore, Billerica, MA, USA). The membranes were incubated with primary antibodies (GAPDH (#60004-1-Ig, Proteintech, Rosemont, IL, USA), SIRT1 (#2028S, Cell Signaling Technology, Danvers, MA, USA), SREBP-1c (#66875-1-Ig, Proteintech, Rosemont, IL, USA), PGC-1$\alpha$ (#NBP1-04676, NOVUS, Stroudsburg, PA, USA), p-AMPK$\alpha$ (#2537S, Cell Signaling Technology, Danvers, MA, USA), AMPK$\alpha$ (#5832S, Cell Signaling Technology, Danvers, MA, USA), ACC (#3676S, Cell Signaling Technology, Danvers, MA, USA), and SCD-1 (#2794S, Cell Signaling Technology, Danvers, MA, USA), and a suitable horseradish-peroxidase (HRP)-conjugated secondary antibody (#ZB-2305, #ZB-2301, ZSGB-BIO, Beijing, China). The protein expression signals were detected by an ECL Western Blotting Substrate (#34094, Thermo Fisher Scientific, Waltham, MA, USA) on a chemiluminescence imaging system (Tanon 5200, Beijing, China). The band intensity was quantified by ImageJ software (National Institutes of Health, Bethesda, MD, USA).

### 4.10. RNA Isolation and Real–Time Quantitative PCR

Total RNA from liver tissues were extracted with TRIzol$^{\text{TM}}$ reagent (1559602, Invitrogen, CA, USA) and reverse-transcribed to cDNA using the *PerfectStart* Uni RT&qPCR Kit (AUQ01, TransGen, Beijing, China). The RT-qPCR assays were performed using *PerfectStart* Green qPCR SuperMix (AQ601, TransGen, Beijing, China) on a FTC3000 RT-qPCR system (Funglyn Biotech Inc, Toronto, ON, Canada). The mRNA expression of the studied genes was calculated relative to *Gapdh*. The primers are listed in Table S1.

### 4.11. Statistical Analysis

Data are presented as mean $\pm$ standard errors of the mean (SEM). GraphPad Prism 8.0 software (GraphPad, San Diego, CA, USA) was used for statistical analysis. The normality and lognormality tests were used for an assumption of the normality of data. The Brown–Forsythe (B-F) test was used for testing the assumption of equal variances. The comparisons among multiple groups were analyzed by one-way analysis of variance (ANOVA) with a Dunnett post-hoc test. If the data did not satisfy the normality, lognormality, and equal variance tests, the results were analyzed further using the Kruskal–Wallis test for multiple comparisons, followed by a Dunnett post-hoc test. $p < 0.05$ was considered statistically significant.

## 5. Conclusions

In summary, our studies elucidated that the SIRT1 activator E1231 decreased the expression of key lipogenesis-related proteins SREBP-1c, ACC and SCD-1 by activating the SIRT1-AMPK$\alpha$ pathway, and thereby inhibited hepatocyte lipid accumulation and improved hepatocyte steatosis, which ultimately had an improvement effect on NAFLD. This study provides new and clear evidence for the development of SIRT1 activators for the treatment of NAFLD.

**Supplementary Materials:** The following supporting information can be downloaded at: https://www.mdpi.com/article/10.3390/cimb45060321/s1, Figure S1: Effects of E1231 on liver inflammatory cytokines in NAFLD mice; Figure S2: E1231 decreased malondialdehyde (MDA) contents in NAFLD mice; Table S1: The primer sequences of the target genes in RT-qPCR assay.

**Author Contributions:** Conceptualization, S.S., Y.X., J.H. and S.L.; methodology, J.H., Y.X. and S.L.; software, J.H.; validation, J.H., S.L., W.W., X.J., C.L., L.L., Y.L., R.S. and Y.Z. (Yuyan Zhang); formal analysis, J.H. and S.L.; investigation, J.H., S.L., W.W., X.J., C.L., L.L., Y.L., R.S., Y.Z. (Yuyan Zhang), J.Z. and Y.Z. (Yuhao Zhang); resources, Y.W.; data curation, J.H. and S.L.; writing—original draft preparation, J.H. and S.L.; writing—review and editing, Y.X. and S.S.; visualization, J.H.; supervision, Y.X. and S.S.; project administration, Y.X. and S.S.; funding acquisition, Y.X. and S.S.; All authors have read and agreed to the published version of the manuscript.

**Funding:** This study was supported by CAMS Innovation Fund for Medical Sciences (CIFMS) (2022-JKCS-10, 2021-1-I2M-030), the Foundation National Natural Science Foundation of China (81973328) and the Chinese Pharmaceutical Association-Yiling Biomedical Innovation Fund (CPA-B04-ZC-2021-005).

**Institutional Review Board Statement:** Not applicable.

**Informed Consent Statement:** Not applicable.

**Data Availability Statement:** Data is available on request from the authors.

**Acknowledgments:** We deeply thank Tingting Feng for her hard work in compound screening and excellent technical support.

**Conflicts of Interest:** The authors declared they do not have anything to disclose regarding the conflict of interest concerning this manuscript.

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
