# Peer review of "SIRT1 Activator E1231 Alleviates Nonalcoholic Fatty Liver Disease by Regulating Lipid Metabolism"

_cimb, doi:10.3390/cimb45060321_

Round 1

Reviewer 1 Report

Comments and Suggestions for Authors

Journal: Current Issues in Molecular Biology

Manuscript ID:  cimb-2375051                         Authors: Jiangxue Han et al.

Manuscript title: "SIRT1 activator E1231 alleviates nonalcoholic fatty liver disease by regulating lipid metabolism"

The authors of the present study tried to explore the potential advantages of the E1231, a SIRT1 activator, in mice with nonalcoholic fatty liver disease (NAFLD) induced by a high-fat and high-cholesterol (HFHC) diet. The study also examined the effects of E1231 on free fatty acid challenged AML12 or HepG2 cells. According to the study findings, E1231 mitigated NAFLD progression in HFHC mice and enhanced liver function by modulating the SIRT1-AMPK pathway. Moreover, the authors suggest that E1231 has the potential to be a therapeutic candidate for NAFLD that warrants further investigation. It is an interesting study that delves into a significant field that requires further exploration of additional therapeutic strategies.

The following comments should be considered.

Comments:

1.     In the last paragraph of the introduction, it would be helpful to emphasize the current gap in knowledge, the study hypothesis, the novelty of the research, and the aims of the study. I would suggest omitting the results and conclusions/future suggestions of the study in this paragraph.

2.     The schedule and design of the animal experiments in Figure 1B could be better explained. For example, the current graph does not indicate how E1231 was administered, which group received it, the size of each group, or whether initial randomization occurred and when the measurements were taken. To help readers better understand the study's schedule of animal experiments, consider providing a new graph that offers additional visual information without being overly detailed. Additionally, provide relevant explanations in the footnote as needed.

3.     In the statistical method, the authors should clarify whether correction for multiple comparisons was performed in the one-way ANOVA analyses. If so, please update the figure legends accordingly. Additionally, did the authors assess normality? Did the authors perform any power analysis?

4.     Did the authors follow the ARRIVE Guidelines? It would be helpful if the authors could complete and submit the ARRIVE Guidelines Checklist with the manuscript.

5.     The concluding sentence in the discussion could be omitted since the authors have included a separate conclusion paragraph at the end of the manuscript.

6.     The conclusion paragraph could be expanded to provide a comprehensive and brief summary of the study's key findings and highlight the exploratory nature of the research and preliminary nature of the findings, along with the potential therapeutic benefits of SIRT1 activators.

7.     In the discussion section, the authors should acknowledge the study's limitations, identify potential areas for future research, and emphasize the importance of further investigation into the therapeutic potential of SIRT1 activators before making any final suggestions.

Reviewer 2 Report

Comments and Suggestions for Authors

Brief Summary

The study by Han et al. investigated the effects of increasing SIRT1 activity via a SIRT1 activator E1231 on NAFLD in a high-fat high-cholesterol diet-fed mouse model as well as FFA-treated cell models. It was concluded that SIRT1 activation can reverse NAFLD potentially through an AMPK-mediated pathway.

The study is overall well designed, executed and presented, demonstrating that a selective SIRT1 activator can be a promising therapeutic treatment option for NAFLD.

General Concept Comments

1.     There are many signaling cross-talks between the liver and white adipose tissue (WAT), and there have been increasing evidences suggesting the importance of WAT metabolism on liver metabolism. There have also been evidences suggesting SIRT1 manipulation in WAT can have an impact on liver metabolism potentially through metabolic substrate delivery or inflammatory signaling changes. It would be helpful for the authors to measure the impact of E1231 on WAT SIRT1 activity, fatty acid metabolism (e.g., lipid content, lipogenesis, lipolysis, morphology, etc.) and inflammation to assess whether changes in WAT play a role in liver metabolic improvements.

Specific Comments

NA

Reviewer 3 Report

Comments and Suggestions for Authors

There are some minor issues that I would like the authors to address:

1. Page 3, Line 5: the authors showed that the 40-week HFHC feeding markedly increased the HDL-C, among other things. Is it normal based on previous studies? Also, how the E1231 treatment did manage to lower other lipid markers and even increase the HDL-C levels? The authors should comment or provide some explanation in the manuscript text.

2. Page 5, Lines 20-25: Authors stated that the mRNA levels of different genes were increased in the HFHC-fed mice; however, the P values associated with these statements are insignificant (more than P=0.05) for some genes in different conditions. They should correct these statements to reflect the P values. In the cases where the P values are not less than 0.05, they can state that "there might be a trend which did not reach statistical significance."

3. Page 10, Line 6: The authors used the EX527, a SIRT1 inhibitor, to show that the impact of E1231 treatment on lipid deposition is through SIRT1. I believe having an EX527 treatment group alone would have been a better experimental control. Can the authors cite more work to show the impact of EX257 treatment in an HFHC-fed or FFA-challenged model?

4. Page 18, Line 1-2: Data should be presented as Mean with SD considering the study design. The comparison between any two groups must be made using an unpaired two-tailed student's t-test (not ANOVA, as the authors stated). Also, the authors must say what post hoc analysis they used after One-way ANOVA when comparing more than two groups.

Round 2

Reviewer 1 Report

Comments and Suggestions for Authors

The authors have satisfactorily responded to my comments and suggestions. They have improved the quality of the paper and made the necessary changes to the manuscript. The revised manuscript is an interesting and well-written paper focusing on an important topic. There are no further considerations.